# Conditional Inference in Pre-trained Variational Autoencoders via Cross-coding

## Abstract

Variational Autoencoders (VAEs) are a popular generative model, but one in which conditional inference can be challenging. If the decomposition into query and evidence variables is fixed, conditional VAEs provide an attractive solution. To support arbitrary queries, one is generally reduced to Markov Chain Monte Carlo sampling methods that can suffer from long mixing times. In this paper, we propose an idea we term *cross-coding* to approximate the distribution over the latent variables after conditioning on an evidence assignment to some subset of the variables. This allows generating query samples *without* retraining the full VAE. We experimentally evaluate three variations of cross-coding showing that (i) they can be quickly optimized for different decompositions of evidence and query and (ii) they quantitatively and qualitatively outperform Hamiltonian Monte Carlo.

## 1 Introduction

Variational Autoencoders (VAEs) (Kingma & Welling, 2014) are a popular deep generative model with numerous extensions including variations for planar flow (Rezende & Mohamed, 2015), inverse autoregressive flow (Kingma et al., 2016), importance weighting (Burda et al., 2016), ladder networks (Maaløe et al., 2016), and discrete latent spaces (Rolfe, 2017) to name just a few. Unfortunately, existing methods for conditional inference in VAEs are limited. Conditional VAEs (CVAEs) (Sohn et al., 2015) allow VAE training conditioned on a fixed decomposition of evidence and query, but are computationally impractical when varying queries are made. Alternatively, Markov Chain Monte Carlo methods such as Hamiltonian Monte Carlo (HMC) (Girolami & Calderhead, 2011; Daniel Levy, 2018) are difficult to adapt to these problems, and can suffer from long mixing times as we show empirically.

To remedy the limitations of existing methods for conditional inference in VAEs, we aim to approximate the distribution over the latent variables after conditioning on an evidence assignment through a variational Bayesian methodology. In doing this, we reuse the decoder of the VAE and show that the error of the distribution over query variables is controlled by that over latent variables via a fortuitous cancellation in the KL-divergence. This avoids the computational expense of re-training the decoder as done by the CVAE approach. We term the network that generates the conditional latent distribution the *cross-coder*.

We experiment with two cross-coding alternatives: Gaussian variational inference via a linear transform (GVI) and Normalizing Flows (NF). We also provide some comparison to a fully connected network (FCN), which suffers from some technical and computational issues but provides a useful point of reference for experimental comparison purposes. Overall, our results show that the GVI and NF variants of cross-coding can be optimized quickly for arbitrary decompositions of query and evidence and compare favorably against a ground truth provided by rejection sampling for low latent dimensionality. For high dimensionality, we observe that HMC often fails to mix despite our systematic efforts to tune its parameters and hence demonstrates poor performance compared to cross-coding in both quantitative and qualitative evaluation.

---

**Algorithm 1** Conditional Inference via Cross-coding.

**Input** (a) Pre-trained VAE $p(\mathbf{z})p_\theta(\mathbf{t}|\mathbf{z})$ with $p_\theta(\mathbf{t}|\mathbf{z})$ based on $\mathrm{Decoder}_\theta(\mathbf{z})$. (Encoder ignored.)
(b) Single query $\mathbf{x}$ (any subset of $\mathbf{t}$) for which to predict $\mathbf{y}$. (Rest of $\mathbf{t}$.)

**Optimize** Define $q(\boldsymbol{\epsilon})q_\psi(\mathbf{z}|\boldsymbol{\epsilon})$ with $q_\psi(\mathbf{z}|\boldsymbol{\epsilon})$ based on $\mathrm{XCoder}_\psi(\boldsymbol{\epsilon})$. Find $\psi$ to maximize C-ELBO$[q_\psi(\mathbf{Z})\|p_\theta(\mathbf{Z},\mathbf{x})]$ (Defined in Theorem 2). Estimate stochastic gradients by drawing random $\boldsymbol{\epsilon} \sim q(\boldsymbol{\epsilon})$ and using the reparameterization trick.

**Predict** Draw a sample $\{\mathbf{z}_m\}_{m=1}^M \sim q_\psi(\mathbf{z})$ by setting $\mathbf{z}_m = \mathrm{XCoder}_\psi(\boldsymbol{\epsilon}_m)$ for $\boldsymbol{\epsilon}_m \sim q(\boldsymbol{\epsilon})$. Predict $p_\theta(\mathbf{y}|\mathbf{x}) \approx \frac{1}{M}\sum_{m=1}^M p_\theta(\mathbf{y}|\mathbf{z}_m)$. (Justified since the optimization phase tightened a bound (Lemma 1) on the divergence between $\int q_\psi(\mathbf{z})p_\theta(\mathbf{y}|\mathbf{z})d\mathbf{z}$ and $p_\theta(\mathbf{y}|\mathbf{x})$ )

---

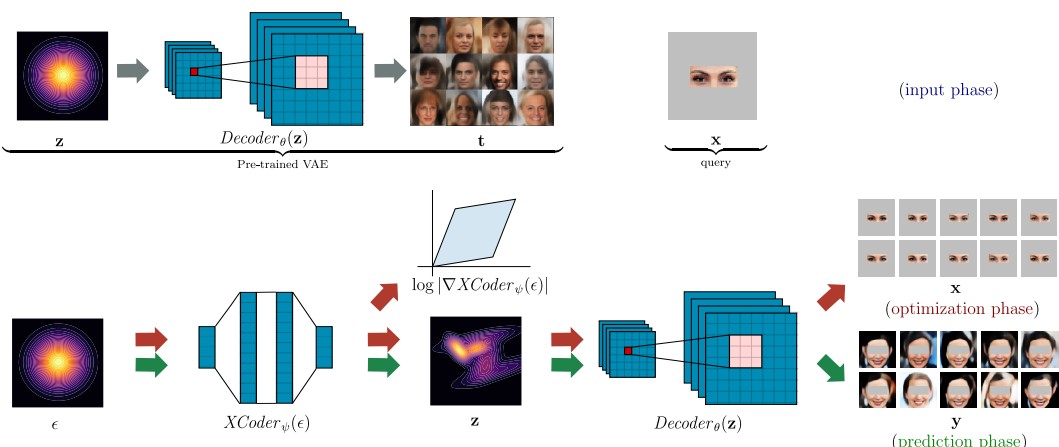

Figure 1: Proposed Cross-coding framework for conditional inference with variational auto-encoders. Arrow and text colors are aligned with the description in Algorithm 1, where $\mathbf{t} = (\mathbf{x}, \mathbf{y})$.

## 2 BACKGROUND

### 2.1 VARIATIONAL AUTO-ENCODERS

One way to define an expressive generative model $p_\theta(\mathbf{t})$ is to introduce latent variables $\mathbf{z}$. Variational Auto-Encoders (VAEs) (Kingma & Welling, 2014) model $p(\mathbf{z})$ as a simple fixed Gaussian distribution. Then, for real $\mathbf{t}$, $p_\theta(\mathbf{t}|\mathbf{z})$ is a Gaussian with the mean determined by a "decoder" network as

$$p_\theta(\mathbf{t}|\mathbf{z}) = \mathcal{N}(\mathbf{t}; \mathrm{Decoder}_\theta(\mathbf{z}), \sigma^2 I). \tag{1}$$

If $\mathbf{t}$ is binary, a product of independent Bernoulli's is parameterized by a sigmoidally transformed decoder. If the decoder network has high capacity, the marginal distribution $p_\theta(\mathbf{t})$ can represent a wide range of distributions. In principle, one might wish to train such a model by (regularized) maximum likelihood. Unfortunately, the marginal $p_\theta(\mathbf{t})$ is intractable. However, a classic idea (Saul et al., 1996) is to use variational inference to lower-bound it. For any distributions $p_\theta$ and $q_\phi$,

$$\log p_\theta(\mathbf{t}) = \log \int_{\mathbf{z}} p_\theta(\mathbf{t},\mathbf{z})d\mathbf{z} = \underbrace{\mathbb{E}_{q_\phi(\mathbf{Z})}\log\frac{p_\theta(\mathbf{Z},\mathbf{t})}{q_\phi(\mathbf{Z})}}_{\mathrm{ELBO}[q_\phi(\mathbf{Z})\|p_\theta(\mathbf{Z},\mathbf{t})]} + KL[q_\phi(\mathbf{Z})\|p_\theta(\mathbf{Z}|\mathbf{t})]. \tag{2}$$

Since the KL-divergence is non-negative, the "evidence lower bound" (ELBO) lower bounds $\log p_\theta(\mathbf{t})$. Thus, as a surrogate to maximizing the likelihood over $\theta$ one can maximize the ELBO over to $\theta$ and $\phi$ simultaneously.

VAEs define $q_\phi(\mathbf{z})$ as the marginal of $q(\mathbf{t})q_\phi(\mathbf{z}|\mathbf{t})$ where $q(\mathbf{t})$ is simple and fixed and $q_\phi(\mathbf{z}|\mathbf{t}) = \mathcal{N}(\mathbf{z}; \mathrm{Encoder}_\phi(\mathbf{t}))$ is a Gaussian with a mean and covariance both determined by an "encoder" network.

## 2.2 THE CONDITIONAL INFERENCE PROBLEM

In this paper, we assume a VAE has been pre-trained. Then, at test time, some arbitrary subset $\mathbf{x}$ of $\mathbf{t}$ is observed as evidence, and the goal is to predict the distribution of the non-observed $\mathbf{y}$ where the decomposition $\mathbf{t} = (\mathbf{x}, \mathbf{y})$ is unpredictable. If this decomposition of $\mathbf{t}$ into evidence and query variables is fixed and known ahead of time, a natural solution is to train an explicit conditional model, the approach taken by Conditional Variational Autoencoders(Sohn et al., 2015). We focus on supporting *arbitrary* queries, where training a conditional model for each possible decomposition $\mathbf{t} = (\mathbf{x}, \mathbf{y})$ is infeasible.

## 3 CONDITIONAL INFERENCE ON VARIATIONAL AUTO-ENCODERS

We now turn to the details of conditional inference. We assume we have pretrained a VAE and now wish to approximate the distribution $p_\theta(\mathbf{y}|\mathbf{x})$ where $\mathbf{x}$ is some new "test" input not known at VAE training time. Unfortunately, exact inference is difficult, since computing this probability exactly would require marginalizing out $\mathbf{z}$.

### 3.1 EXPLOITING FACTORIZATION IN THE OUTPUT

One helpful property comes from the fact that in a VAE, the conditional distribution over the output (Eq. 1) has a diagonal covariance, which leads to the following decomposition:

**Observation 1** The distribution of a VAE can be factorized as $p_\theta(\mathbf{x}, \mathbf{y}, \mathbf{z}) = p(\mathbf{z})p_\theta(\mathbf{x}|\mathbf{z})p_\theta(\mathbf{y}|\mathbf{z})$.

Since $\mathbf{x}$ and $\mathbf{y}$ are conditionally independent given $\mathbf{z}$, the conditional of $\mathbf{y}$ given $\mathbf{x}$ can be written as

$$p_\theta(\mathbf{y}|\mathbf{x}) = \int_\mathbf{z} p_\theta(\mathbf{z}, \mathbf{y}|\mathbf{x})p_\theta d\mathbf{z} = \int_\mathbf{z} p_\theta(\mathbf{z}|\mathbf{x})p_\theta(\mathbf{y}|\mathbf{z})d\mathbf{z}. \tag{3}$$

Here, $p_\theta(\mathbf{y}|\mathbf{z})$ can easily be evaluated or simulated. However $p_\theta(\mathbf{z}|\mathbf{x})$ is much more difficult to work with since it involves "inverting" the decoder. This factorization can also be exploited by Markov chain Monte Carlo methods (MCMC), such as Hamiltonian Monte Carlo (HMC) (Girolami & Calderhead, 2011; Daniel Levy, 2018). In this case, it allows the Markov chain to be defined over $\mathbf{z}$ alone, rather than $\mathbf{z}$ and $\mathbf{y}$ together. That is, one can use MCMC to attempt sampling from $p_\theta(\mathbf{z}|\mathbf{x})$, and then draw exact samples from $p_\theta(\mathbf{y}|\mathbf{z})$ just by evaluating the decoder network at each of the samples of $\mathbf{z}$. The experiments using MCMC in Section 4 use this strategy.

### 3.2 VARIATIONAL INFERENCE BOUNDS

The basic idea of variational inference (VI) is to posit some distribution $q_\psi$, and optimize $\psi$ to make it match the target distribution as closely as possible. So, in principle, the goal of VI would be to minimize $KL[q_\psi(\mathbf{Y})\|p_\theta(\mathbf{Y}|\mathbf{x})]$. For an arbitrary distribution $q_\psi$ this divergence would be difficult to work with due to the need to marginalize out $\mathbf{z}$ in $p_\theta$ as in Eq. 3.

However, if $q_\psi$ is chosen carefully, then the above divergence can be upper-bounded by one defined directly over $\mathbf{Z}$. Specifically, we will choose $q_\psi$ so that the dependence of $\mathbf{y}$ on $\mathbf{z}$ under $q_\psi$ is the same as under $p_\theta$ (both determined by the "decoder").

**Lemma 1.** *Suppose we choose $q_\psi(\mathbf{z}, \mathbf{y}) = q_\psi(\mathbf{z})p_\theta(\mathbf{y}|\mathbf{z})$. Then*

$$KL[q_\psi(\mathbf{Y})\|p_\theta(\mathbf{Y}|\mathbf{x})] \leq KL[q_\psi(\mathbf{Z})\|p_\theta(\mathbf{Z}|\mathbf{x})]. \tag{4}$$

This is proven in the Appendix. The result follows from using the chain rule of KL-divergence (Cover & Thomas, 2006) to bound the divergence over $\mathbf{y}$ by the divergence jointly over $\mathbf{y}$ and $\mathbf{z}$. Then the common factors in $q_\psi$ and $p_\theta$ mean this simplifies into a divergence over $\mathbf{z}$ alone.

Given this Lemma, it makes sense to seek a distribution $q_\psi$ such that the divergence on the right-hand side of Eq. 4 is as low as possible. To minimize this divergence, consider the decomposition

$$\log p_\theta(\mathbf{x}) = \underbrace{\mathbb{E}_{q_\phi(\mathbf{Z})} \log \frac{p_\theta(\mathbf{Z}, \mathbf{x})}{q_\psi(\mathbf{Z})}}_{\text{C-ELBO}[q_\psi(\mathbf{Z})\|p_\theta(\mathbf{Z}, \mathbf{x})]} + KL[q_\psi(\mathbf{Z})\|p_\theta(\mathbf{Z}|\mathbf{x})], \tag{5}$$

which is analogous to Eq. 2. Here, we call the first term the "conditional ELBO" (C-ELBO) to reflect that maximizing it is equivalent to minimizing an upper bound on $KL[q_\psi(\mathbf{Y})\|p_\theta(\mathbf{Y}|\mathbf{x})]$.

## 3.3 Inference via Cross-coding

The previous section says that we should seek a distribution $q_\psi$ to approximate $p_\theta(\mathbf{z}|\mathbf{x})$. Although the latent distribution $p(\mathbf{z})$ may be simple, the conditional distribution $p_\theta(\mathbf{z}|\mathbf{x})$ is typically complex and often multimodal (cf. Fig. 3).

To define a variational distribution satisfying the conditions of Lemma 1, we propose to draw $\epsilon$ from some fixed base density $q(\epsilon)$ and then use a network with parameters $\psi$ to map to the latent space $\mathbf{z}$ so that the marginal $q_\psi(\mathbf{z})$ is expressive. The conditional of $\mathbf{y}$ given $\mathbf{z}$ is exactly as in $p$. The full variational distribution is therefore

$$q_\psi(\epsilon, \mathbf{z}, \mathbf{y}) = q(\epsilon)q_\psi(\mathbf{z}|\epsilon)p_\theta(\mathbf{y}|\mathbf{z}) \qquad \text{with} \qquad q_\psi(\mathbf{z}|\epsilon) = \delta(\mathbf{z} - \text{XCoder}_\psi(\epsilon)), \qquad (6)$$

where $\delta$ is a multivariate delta function. We call this network a "Cross-coder" to emphasize that the parameters $\psi$ are fit so that $q_\psi(\mathbf{Z})$ matches $p_\theta(\mathbf{Z}|\mathbf{x})$, and so that $\mathbf{z}$, when "decoded" using $\theta$, will predict $\mathbf{y}$ given $\mathbf{x}$.

**Theorem 2.** *If $q_\psi$ is as defined in Eq. 6 and $\text{XCoder}_\psi(\epsilon)$ is one-to-one for all $\psi$, the C-ELBO from Eq. 5 becomes*

$$\text{C-ELBO}[q_\psi(\mathbf{Z})\|p_\theta(\mathbf{Z}, \mathbf{x})] = \mathbb{E}_{q(\epsilon)}\left[\log p_\theta(\text{XCoder}_\psi(\epsilon), \mathbf{x}) + \log|\nabla\text{XCoder}_\psi(\epsilon)|\right] + \mathbb{H}[q(\epsilon)],$$

*where $\mathbb{H}[q(\epsilon)]$ is the (fixed) entropy of $q(\epsilon)$, $\nabla$ is the Jacobian with respect to $\epsilon$, and $|\cdot|$ is the determinant.*

Informally, this result can be proven as follows: the C-ELBO was defined on $\mathbf{z}$ alone, while our definition of $q_\psi$ in Eq. 6 also involves $\mathbf{y}$ and $\epsilon$. Marginalizing out $\mathbf{y}$ is trivial. Then, since $\mathbf{z}$ and $\epsilon$ are deterministically related under $q_\psi$ one can change variables to convert the expectation over $\mathbf{z}$ to one over $\epsilon$, leaving the log-determinant Jacobian as an artifact of the entropy of $q_\psi(\mathbf{z})$.

This objective is related to the "triple ELBO" used by Vedantam et al. (2017) for a situation with a small number of *fixed* decompositions of $\mathbf{t}$ into $(\mathbf{x}, \mathbf{y})$. Algorithmically, the approaches are quite different since they pre-train a single network for each subset of $\mathbf{t}$, which can be used for any $\mathbf{x}$ with that pattern, and a futher product of experts approximation is used for novel missing features at test time. We assume arbitrary queries and so pre-training is inapplicable and novel missing features pose no issue. Still, our bounding justification may provide additional insight for their approach.

## 3.4 Cross-Coders

We explore the following two candidate Cross-Coders.

**Gaussian Variational Inference (GVI):** The GVI $\text{XCoder}_\psi$ linearly warps a spherical Gaussian over $\epsilon$ into an arbitrary Gaussian $\mathbf{z}$:

$$\text{XCoder}_\psi(\epsilon) = \mathbf{W}\epsilon + \mathbf{b}, \qquad \text{where} \qquad \log|\nabla\text{XCoder}_\psi(\epsilon)| = \log|\mathbf{W}|, \qquad (7)$$

where $\psi = (\mathbf{W}, \mathbf{b})$ for a square matrix $\mathbf{W}$ and a mean vector $\mathbf{b}$. While projected gradient descent can be used to maintain invertibility of $W$, we did not encounter issues with non-invertible $W$ requiring projection during our experiments.

**Normalizing Flows (NF):** A normalizing flow (Rezende & Mohamed, 2015) projects a probability density through a sequence of easy computable and invertible mappings. By stacking multiple mappings, the transformation can be complex. We use the special structured network called Planar Normalizing Flow:

$$\mathbf{h}_i = f_i(\mathbf{h}_{i-1}) = \mathbf{h}_{i-1} + \mathbf{u}_i g(\mathbf{h}_{i-1}^T\mathbf{w}_i + b_i), \qquad (8)$$

for all $i$, where $h_0 = \epsilon$, $i$ is the layer id, $w$ and $u$ are vectors, and the output dimension is exactly same with the input dimension. Using $\circ$ for function composition, the $\text{XCoder}_\psi$ is given as

$$\text{XCoder}_\psi(\epsilon) = f_k \circ f_{k-1} \cdots f_1(\epsilon), \qquad \text{where} \qquad \log|\nabla\text{XCoder}_\psi(\epsilon)| = \sum_{i=1}^{k}\log|\nabla f_i|. \qquad (9)$$

The bound in Theorem 2 requires that $\text{XCoder}_\psi$ is invertible. Nevertheless, we find **Fully Connected Networks (FCNs)** useful for comparison in low-dimensional visualizations. Here, the Jacobian must be calculated using separate gradient calls for each ouput variable, and the lack of invertibility prevents the C-ELBO bound from being correct.

We summarize our approach in Algorithm 1. In brief, we define a variational distribution $q_\psi(\epsilon, \mathbf{z}) = q(\epsilon)q_\psi(\mathbf{z}|\epsilon)$ and optimize $\psi$ so that $q_\psi(\mathbf{z})$ is close to $p_\theta(\mathbf{z}|\mathbf{x})$. The variational distribution includes a "CrossCoder" as $q_\psi(\mathbf{z}|\epsilon) = \delta(\mathbf{z} - \text{XCoder}_\psi(\epsilon))$. The algorithm uses stochastic gradient decent on the C-ELBO with gradients estimated using Monte Carlo samples of $\epsilon$ and the reparameterization trick (Kingma & Welling, 2014; Titsias & Lázaro-Gredilla, 2014; Rezende et al., 2014). After inference, the original VAE distribution $q(\mathbf{y}|\mathbf{z}) = p_\theta(\mathbf{y}|\mathbf{z})$ gives samples over the query variables.

## 4 EXPERIMENTS

Having defined our cross-coding methodology for conditional inference with pre-trained VAEs, we now proceed to empirically evaluate our three previously defined XCoder instantiations and compare them with (Markov chain) Monte Carlo (MCMC) sampling approaches on three different pre-trained VAEs. Below we discuss our datasets and methodology followed by our experimental results.

### 4.1 DATASETS AND PRE-TRAINED VAEs

**MNIST** is the well-known benchmark handwritten digit dataset (LeCun & Cortes, 2010). We use a pre-trained VAE with a fully connected encoder and decoder each with one hidden layer of 64 ReLU units, a final sigmoid layer with Bernoulli likelihood, and 2 latent dimensions for $\mathbf{z}$.[1] The VAE has been trained on 60,000 black and white binary thresholded images of size $28 \times 28$. The limitation to 2 dimensions allows us to visualize the conditional latent distribution of all methods and compare to the ground truth through a fine-grained discretization of $\mathbf{z}$.

**Anime** is a dataset of animated character faces (Jin et al., 2017). We use a pre-trained VAE with convolutional encoder and deconvolutional decoder, each with 4 layers. The decoder contains respective channel sizes $(256, 128, 32, 3)$ each using $5 \times 5$ filters of stride 2 and ReLU activations followed by batch norm layers. The VAE has a final tanh layer with Gaussian likelihood, and 64 latent dimensions for $\mathbf{z}$.[2] The VAE has been trained on 20000 images encoded in RGB of size $64 \times 64 \times 3$.

**CelebA** dataset (Liu et al., 2015) is a benchmark dataset of images of celebrity faces. We use a pre-trained VAE with a structure that exactly matches the Anime VAE provided above, except that it uses 100 latent dimensions for $\mathbf{z}$.[3] The VAE has been trained on 200,000 images encoded in RGB of size $64 \times 64 \times 3$.

### 4.2 METHODS COMPARED

For sampling approaches, we evaluate rejection sampling **(RS)**, which is only feasible for our MNIST VAE with a 2-dimensional latent embedding for $\mathbf{z}$. We also compare to the MCMC method of Hamiltonian Monte Carlo **(HMC)** (Girolami & Calderhead, 2011; Daniel Levy, 2018). Both sampling methods exploit the VAE decomposition and sampling methodology described in Section 3.1.

We went to great effort to tune the parameters of HMC. For MNIST, with low dimensions, this was generally feasible, with a few exceptions as noted in Figure 4(b). For the high-dimensional latent space of the Anime and CelebA VAEs, finding parameters to achieve good mixing was often impossible, leading to poor performance. Section 6.4 of the Appendix discusses this in detail.

For the cross-coding methods, we use the three XCoder variants described in Section 3.3: Gaussian Variational Inference **(GVI)**, Planar Normalizing Flow **(NF)**, and a Fully Connected Neural Network **(FCN)**. By definition, the latent dimensionality of $\epsilon$ must match the latent dimensionality of $\mathbf{z}$ for each pre-trained VAE. Given evidence as described in the experiments, all cross-coders were trained as described in Algorithm 1. We could not train the FCN XCoder for conditional inference in Anime and

---

[1] https://github.com/kvfrans/variational-autoencoder
[2] URL for pre-trained VAE suppressed for anonymous review.
[3] https://github.com/yzwxx/vae-celebA

CelebA due to the infeasibility of computing the Jacobian for the respective latent dimensionalities of these two VAEs.

In preliminary experiments, we considered the alternating sampling approach suggested by (Rezende et al., 2014, Appendix F), but found it to perform very poorly when the evidence is ambiguous. We provide a thorough analysis of this in Section 6.3 of the Appendix comparing results on MNIST with various fractions of the input taken as evidence. In summary, Rezende's alternation method produces reasonable results when a large fraction of pixels are observed, so the posterior is highly concentrated. When less than around 40% are observed, however, performance rapidly degrades.

### 4.3 Evaluation Methodology

We experiment with a variety of evidence sets to demonstrate the efficiency and flexibility of our cross-coding methodology for arbitrary conditional inference queries in pre-trained VAEs. All cross-coding optimization and inference takes (typically well) under 32 seconds per evidence set for all experiments running on an Intel Xeon E5-1620 v4 CPU with 4 cores, 16Gb of RAM, and an NVIDIA GTX1080 GPU. A detailed running time comparison is provided in Section 6.5 of the Appendix.

Qualitatively, we visually examine the 2D latent distribution of $\mathbf{z}$ conditioned on the evidence for the special case of MNIST, which has low enough latent dimensionality to enable us to obtain ground truth through discretization. For all experiments, we qualitatively assess sampled query images generated for each evidence set to assess both the coverage of the distribution and the quality of match between the query samples and the evidence, which is fixed in the displayed images.

Quantitatively, we evaluate the performance of the proposed framework and candidate inference methods through the following two metrics.

**C-ELBO:** As a comparative measure of inference quality for each of the XCoder methods, we provide pairwise scatterplots of the C-ELBO as defined in 5 for a variety of different evidence sets

**Query Marginal Likelihood:** For each conditional inference evaluation, we randomly select an image and then a subset of that image as evidence $\mathbf{x}$ and the remaining pixels $\mathbf{y}$ as the ground truth query assignment. Given this, we can evaluate the marginal likelihood of the query $\mathbf{y}$ as follows:

$$\log p(\mathbf{y}) = \log E_{\mathbf{z}}[p(\mathbf{y}|\mathbf{Z})]$$

### 4.4 Conditional Inference on MNIST

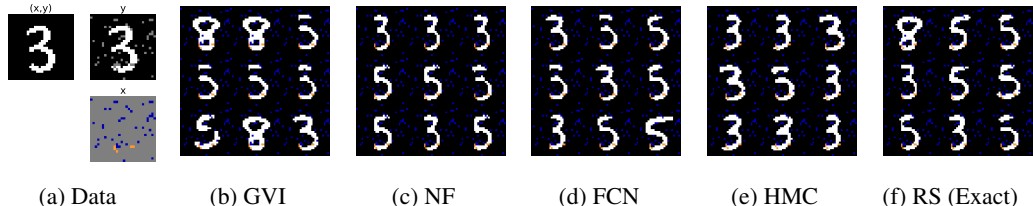

| (a) Data | (b) GVI | (c) NF | (d) FCN | (e) HMC | (f) RS (Exact) |

Figure 2: One conditional inference example for MNIST. In all plots, the evidence subset has white replaced with orange and black replaced with blue. (a) The original digit $\mathbf{t}$, the subset selected for evidence $\mathbf{x}$, and the remaining ground truth query $\mathbf{y}$. (b–f) Nine sample queries from method.

For conditional inference in MNIST, we begin with Figure 2, which shows one example of conditional inference in the pre-trained MNIST model using the different inference methods. While the original image used to generate the evidence represents the digit 3, the evidence is very sparse allowing the plausible generation of other digits. It is easy to see that most of the methods can handle this simple conditional inference, with only GVI producing some samples that do not match the evidence well in this VAE with 2 latent dimensions.

To provide additional insight into Figure 2, we now turn to Figure 3, where we visually compare the true conditional latent distribution $p(\mathbf{z}|\mathbf{x})$ (leftmost) with the corresponding distributions of each of the inference methods. At a first glance, we note that the true distribution is both multimodal and non-Gaussian. We see that GVI covers some mass not present in the true distribution that explains its

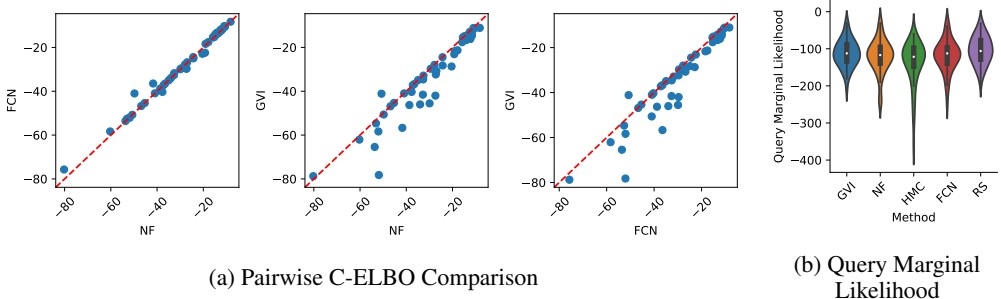

Figure 3: $p(\mathbf{z}|\mathbf{x})$ for the MNIST example in Figure 2. The contour plot (left) shows the true distribution. The remaining plots show samples from each method overlaid on the true distribution.

relatively poor performance in Figure 2(b). All remaining methods (both XCoder and sampling) do a reasonable job of covering the irregular shape and mass of the true distribution.

(a) Pairwise C-ELBO Comparison

(b) Query Marginal Likelihood

Figure 4: (a) Pairwise C-ELBO comparison of different XCoder methods evaluated over the 50 randomly generated evidence sets for MNIST. (b) Violin (distribution) plots of the Query Marginal Likelihood for the same 50 evidence sets from (a), with each likelihood expectation generated from 500 samples. *For both metrics, higher is better*.

We now proceed to a quantitative comparison of performance on MNIST over 50 randomly generated queries. In Figure 4(a), we present a pairwise comparison of the performance of each XCoder method on 50 randomly generated evidence sets. Noting that higher is better, we observe that FCN and NF perform comparably and generally outperform GVI. In Figure 4(b), we examine the Query Marginal Likelihood distribution for the same 50 evidence sets from (a) with each likelihood expectation generated from 500 samples. Again, noting that higher is better, here we see that RS slightly edges out all other methods with all XCoders generally performing comparably. HMC performs worst here, where we remark that inadequate coverage of the latent $\mathbf{z}$ due to poor mixing properties leads to over-concentration on $\mathbf{y}$ leading to a long tail in a few cases with poor coverage. We will see that these issues with HMC mixing become much more pronounced as we move to experiments in VAEs with higher latent dimensionality in the next section.

## 4.5 CONDITIONAL INFERENCE ON ANIME AND CELEBA

Now we proceed to our larger VAEs for Anime and CelebA with respective latent dimensionality of 64 and 100 that allow us to work with larger and more visually complex RGB images. In these cases, FCN could not be applied due to the infeasibilty of computing the Jacobian and RS is also infeasible for such high dimensionality. Hence, we only compare the two XCoders GVI and NF with HMC.

We now continue to a qualitative and quantitative performance analysis of conditional inference for the Anime and CelebA VAEs. Qualitatively, in Figure 5 for Anime, we see that inference for both the NF XCoder and HMC show little identifiable variation and seem to have collapsed into a single latent mode. In contrast, GVI appears to show better coverage, generating a wide range of faces that generally match very well with the superimposed evidence. For Figure 6, HMC still performs poorly, but NF appears to perform much better, with both XCoders GVI and NF generating a wide range of faces that match the superimposed evidence, with perhaps slightly more face diversity for GVI.

Quantitatively, Figure 7 strongly reflects the qualitative visual observations above. In short for the XCoders, GVI solidly outperforms NF on the C-ELBO comparison. For all methods evaluated on

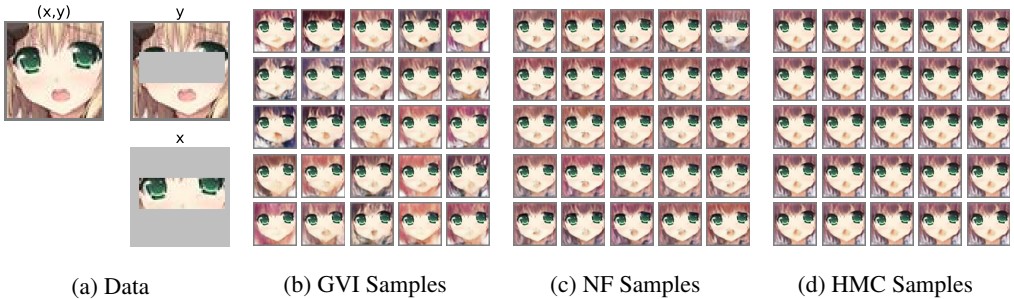

(a) Data        (b) GVI Samples        (c) NF Samples        (d) HMC Samples

Figure 5: One conditional inference example for Anime. (a) The original image $\mathbf{t}$, the subset selected for evidence $\mathbf{x}$, and the remaining ground truth query $\mathbf{y}$. (b–d) 25 sample queries from each method with the evidence superimposed on each image. (c,d) NF and HMC demonstrate poor coverage.

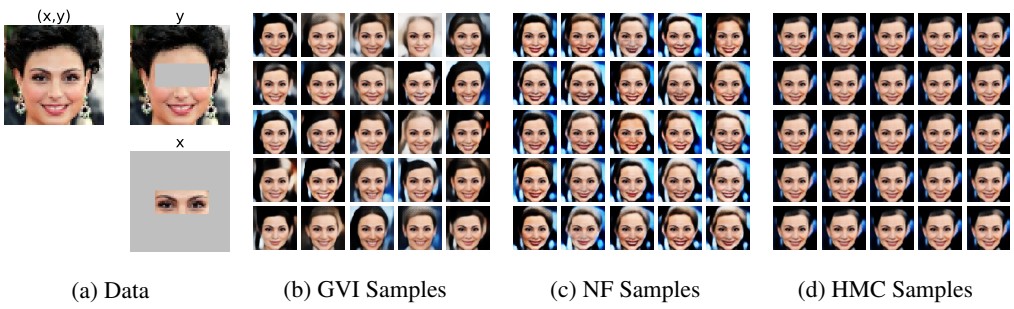

(a) Data        (b) GVI Samples        (c) NF Samples        (d) HMC Samples

Figure 6: One conditional inference example for CelebA. (a) The original image $\mathbf{t}$, the subset selected for evidence $\mathbf{x}$, and the remaining ground truth query $\mathbf{y}$. (b–d) 25 sample queries from each method with the evidence superimposed on each image. (d) HMC demonstrates poor coverage.

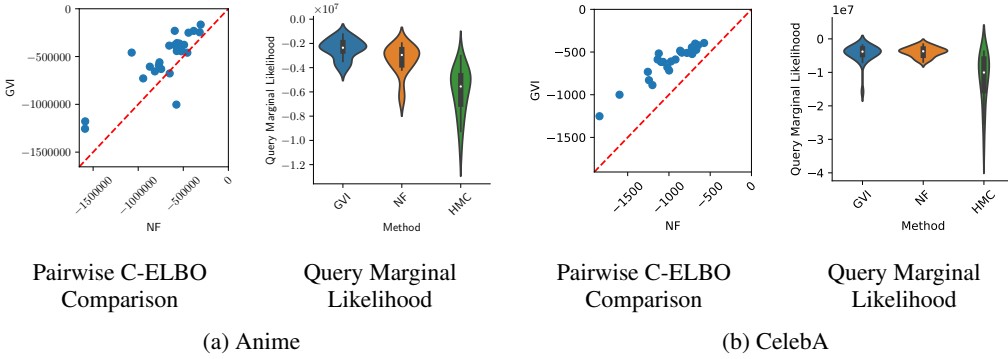

| Pairwise C-ELBO Comparison | Query Marginal Likelihood | Pairwise C-ELBO Comparison | Query Marginal Likelihood |

(a) Anime        (b) CelebA

Figure 7: (left)(a,b) Pairwise C-ELBO comparison of GVI vs. FCN and (right)(a,b) Violin (distribution) plots of the Query Marginal Likelihood for (a) Anime and (b) CelebA. Evaluation details match those of Fig. 4 except with 25 conditional inference queries. *For both metrics, higher is better.*

Query Marginal Likelihood, GVI outperforms both NF and HMC on Anime, while for CelebA GVI performs comparably to (if not slightly worse) than NF, with both solidly outperforming HMC.

## 5    CONCLUSION

We introduced Cross-coding, a novel variational inference method for conditional queries in pre-trained VAEs that does not require retraining the decoder. Using three VAEs pre-trained on different datasets, we demonstrated that the Gaussian Variational Inference (GVI) and Normalizing Flows (NF) cross-coders generally outperform Hamiltonian Monte Carlo both qualitatively and quantitively, thus providing a novel and efficient tool for conditional inference in VAEs with arbitrary queries.

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

## 6 APPENDIX

### 6.1 PROOFS

*Proof Of Lemma 1.* To show this, we first note that the joint divergence over $\mathbf{Y}$ and $\mathbf{Z}$ is equivalent to one over $\mathbf{Z}$ only.

$$
\begin{aligned}
KL[q_\psi(\mathbf{Y}, \mathbf{Z}) \| p_\theta(\mathbf{Y}, \mathbf{Z}|\mathbf{x})] &= KL[q_\psi(\mathbf{Z}) \| p_\theta(\mathbf{Z}|\mathbf{x})] + KL[q_\psi(\mathbf{Y}|\mathbf{Z}) \| p_\theta(\mathbf{Y}|\mathbf{Z}, \mathbf{x})] \\
&\quad \text{by the chain rule of KL-divergence} \\
&= KL[q_\psi(\mathbf{Z}) \| p_\theta(\mathbf{Z}|\mathbf{x})] + KL[q_\psi(\mathbf{Y}|\mathbf{Z}) \| p_\theta(\mathbf{Y}|\mathbf{Z})] \\
&\quad \text{since } \mathbf{Y} \perp \mathbf{X} | \mathbf{Z} \text{ in both } q_\psi \text{ and } p_\theta \\
&= KL[q_\psi(\mathbf{Z}) \| p_\theta(\mathbf{Z}|\mathbf{x})] \\
&\quad \text{since } q_\psi(\mathbf{y}|\mathbf{z}) = p_\theta(\mathbf{y}|\mathbf{z})
\end{aligned}
$$

Then, the result follows just from observing (again by the chain rule of KL-divergence) that

$$
KL[q_\psi(\mathbf{Y}) \| p_\theta(\mathbf{Y}|\mathbf{x})] \le KL[q_\psi(\mathbf{Y}, \mathbf{Z}) \| p_\theta(\mathbf{Y}, \mathbf{Z}|\mathbf{x})].
$$

□

*Proof of Theorem 2.* For the purpose of this proof, use $c_\psi$ to denote CrossEncoder$_\psi$. Firstly, note that the marginal density of $q_\psi(\mathbf{z}|\mathbf{x})$ is (via the standard formula for a change of variables (Kaplan, 1952))

$$
q_\psi(\mathbf{z} = c_\psi(\boldsymbol{\epsilon})) \, |\nabla c_\psi(\boldsymbol{\epsilon})| = q(\boldsymbol{\epsilon})
$$

Thus, we can write

$$
\begin{aligned}
\text{C-ELBO}[q_\psi(\mathbf{Z}) \| p_\theta(\mathbf{Z}, \mathbf{x})] &= \mathbb{E}_{q_\psi(\mathbf{Z})} \log \frac{p_\theta(\mathbf{Z}, \mathbf{x})}{q_\psi(\mathbf{Z})} \\
&= \mathbb{E}_{q(\boldsymbol{\epsilon})} \log \frac{p_\theta(\mathbf{Z} = c_\psi(\boldsymbol{\epsilon}), \mathbf{x})}{q_\psi(\mathbf{Z} = c_\psi(\boldsymbol{\epsilon}))} \\
&= \mathbb{E}_{q(\boldsymbol{\epsilon})} \log \frac{p_\theta(\mathbf{Z} = c_\psi(\boldsymbol{\epsilon}), \mathbf{x})}{q(\boldsymbol{\epsilon}) / |\nabla c_\psi(\boldsymbol{\epsilon})|} \\
&= \mathbb{E}_{q(\boldsymbol{\epsilon})} \left[ \log p_\theta(c_\psi(\boldsymbol{\epsilon}), \mathbf{x}) + \log |\nabla c_\psi(\boldsymbol{\epsilon})| \right] + \mathbb{H}_q[\boldsymbol{\epsilon}].
\end{aligned}
$$

□

### 6.2 PRELIMINARY CHECK OF INFERENCE METHODS

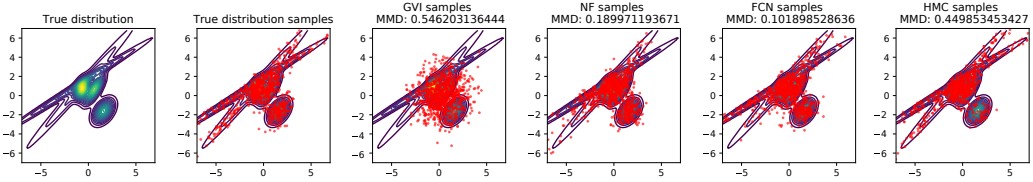

Figure 8: Comparison of different inference methods on modeling a Gaussian mixture model distribution. The true distribution samples are directly sampled from a Gaussian mixture model. Maximum mean discrepancy (MMD) values given in the plot titles are generated relative to the true sample distribution.

In this experiment, we do not use a VAE, but instead simply model a complex latent 2D multimodal distribution over $\mathbf{z}$ as a Gaussian mixture model to evaluate the ability of each conditional inference method to accurately draw samples from this complex distribution. In general, Figure 8 shows that while the XCoders NF and FCN work well here, GVI (by definition) cannot model this multimodal distribution and HMC draws too few samples from the disconnected mode compared to the true sample distribution, indicating slight failure to mix well.

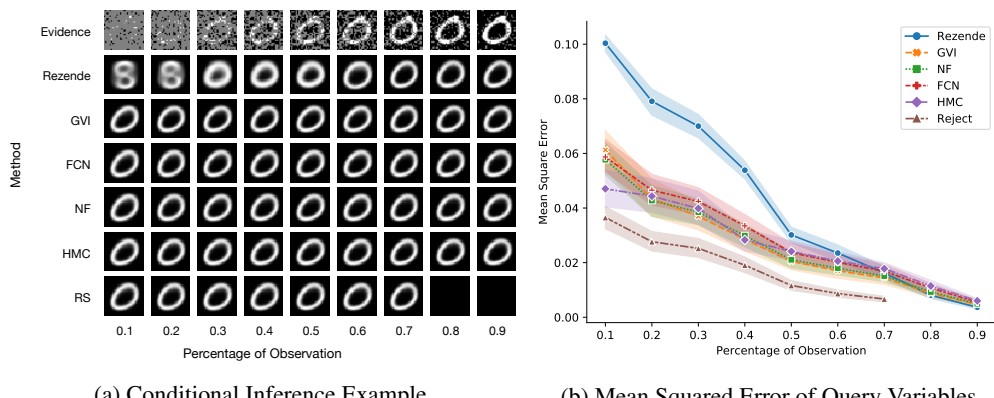

(a) Conditional Inference Example  (b) Mean Squared Error of Query Variables

Figure 9: Comparison of different conditional inference methods include the Rezende method on the MNIST dataset. (a) Shows one intuitive example. The first row shows the evidence observed, and the following rows show the mean of generated samples from the different algorithms. We note that with very high evidence, the posterior becomes extremely concentrated, meaning the rejection rates for rejection sampling become impractical. (b) The mean squared error between query variables of the original image and the generated samples of different algorithms. The results and standard deviations at each observation percentage come from 50 independent randomly selected queries.

### 6.3 Comparison to Rezende Alternation

We compare to the alternating sampling approach of Rezende et al. (2014) (Appendix Section F) which is essentially an approximation of block Gibbs sampling. We call it the "Rezende method" in the following. This method does not asymptotically sample from the conditional distribution since the step sampling the latent variables given the query variables are approximated using the encoder.

Figure 9(a) shows one experiment comparing all candidate algorithms including the Rezende method. We noticed that it fails to generate images that match the evidence when less than 40% of pixels are observed as evidence, while it makes reasonable predictions when the observation rate is higher. Figure 9(b) shows this result is consistent over 50 randomly selected queries.

### 6.4 Systematic HMC Tuning Analysis for Anime and CelebA

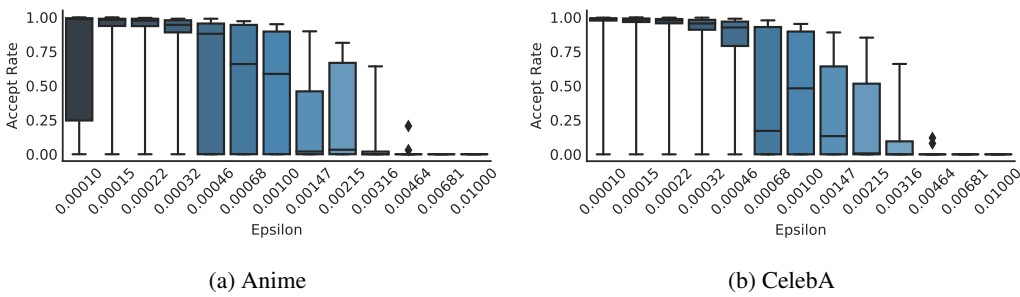

(a) Anime  (b) CelebA

Figure 10: Boxplots of acceptance rate distribution of HMC for 30 Markov Chains vs different $\epsilon$ on (a) Anime and (b) CelebA. Each Markov chain ran for 10,000 burn-in samples with 10 leapfrog steps per iteration.

While tuning HMC in lower dimensions was generally feasible for MNIST with a few exceptions noted in previous discussion of Figure 4(b), we observed that HMC becomes very difficult to tune in the Anime and CelebA VAEs with higher latent dimensionality. To illustrate these HMC tuning difficulties, we present a summary of our systematic efforts to tune HMC on Anime and CelebA in Figure 10 with boxplots of the acceptance rate distribution of HMC for 30 Markov Chains vs different $\epsilon$ on (a) Anime and (b) CelebA. We ran each Markov chain for 10,000 burn-in samples

with 10 leapfrog steps per iteration; we tried 3 different standard leapfrog step settings of $\{5, 10, 30\}$, finding that 10 leapfrog steps provided the best performance across a range of $\epsilon$ and hence chosen for Figure 10.

In short, Figure 10 shows that only a very narrow band of $\epsilon$ lead to a reasonable acceptance rate for good mixing properties of HMC. Even then, however, the distribution of acceptance rates for any particular Markov Chain for a good $\epsilon$ is still highly unpredictable as given by the quartile ranges of the boxplot. In summary, we found that despite our systematic efforts to tune HMC for higher dimensional problems, it was difficult to achieve a good mixing rate and overall contributes to the generally poor performance observed for HMC on Anime and CelebA that we discuss next.

## 6.5 COMPARISON OF RUNNING TIME

The running time of conditional inference with XCoding varies with the complexity of XCoders, the optimization algorithm used, and the complexity of the pretrained Decoder. We found that L-BFGS(Liu & Nocedal, 1989) consistently converged fastest and with the best results in comparison to SGD, Adam, Adadelta, and RMSProp. Table 1, which follows, shows the computation time for each of the three candidate XCoders (FCN is only applicable to MNIST) as well as HMC and Rejection Sampling (RS is only applicable for MNIST).

Table 1: Average running Time (in seconds) of experiments. We use L-BFGS for XCoders in this table. For HMC, we predefine the burn-in (optimization) iterations to be 1000 for all datasets. For all methods, the sample size is 500.

| Period | MNIST | | | | | Amine | | | CelebA | | |
|---|---|---|---|---|---|---|---|---|---|---|---|
| - | GVI | NF | FCN | HMC | RS | GVI | NF | HMC | GVI | NF | HMC |
| Optimization | 0.36 | 2.79 | 5.26 | 34.92 | - | 2.52 | 4.22 | 81.3 | 31.45 | 9.95 | 224.5 |
| Prediction | < 0.04 | < 0.04 | < 0.04 | 2 | 0.19 | < 0.08 | < 0.08 | 2 | < 0.37 | < 0.37 | 2 |

## 6.6 QUALITY OF THE PRE-TRAINED VAE MODELS

To assess the quality of the pre-trained VAE models, we show 100 samples from each in Figure 11.

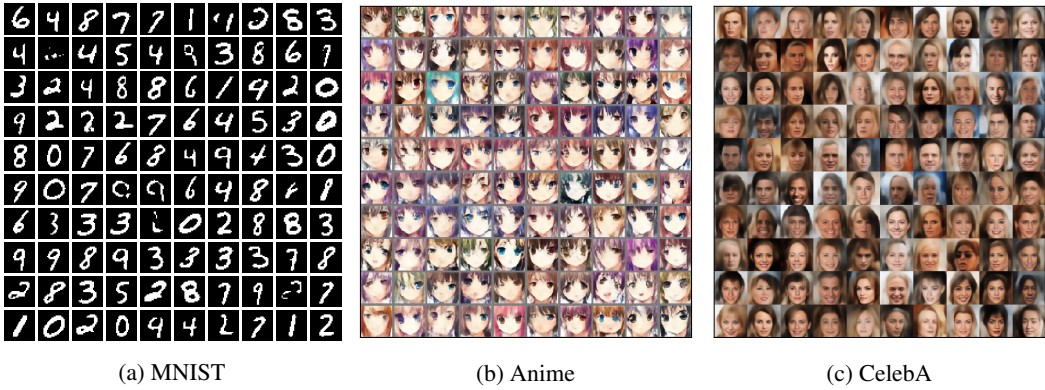

(a) MNIST  (b) Anime  (c) CelebA

Figure 11: Samples from each of the pre-trained VAE models.

## 6.7 MORE INFERENCE EXAMPLES

In Figures 12 and 13, we show two additional examples of conditional inference matching the structure of experiments shown in Figures 5 and 6 in the main text. Overall, we observe the same general trends as discussed in the main text for Figures 5 and 6.

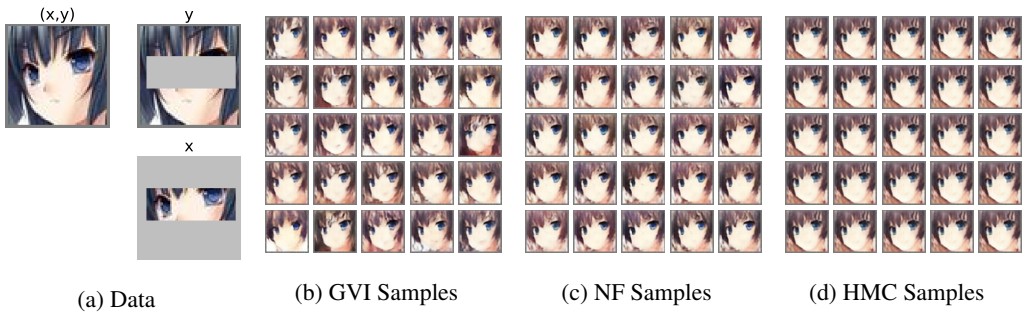

(a) Data      (b) GVI Samples      (c) NF Samples      (d) HMC Samples

Figure 12: Another conditional inference example on Anime dataset

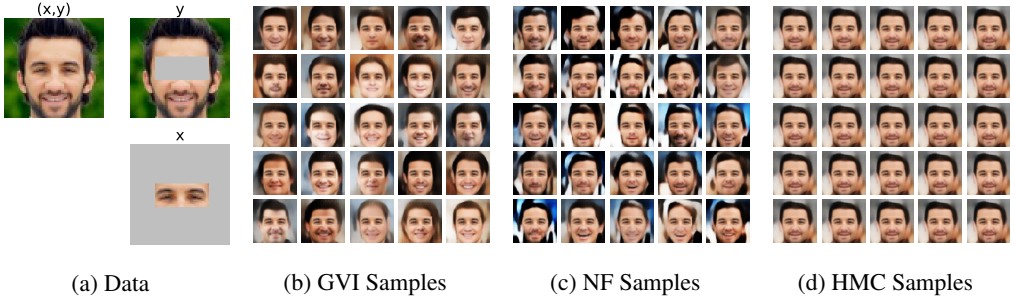

(a) Data      (b) GVI Samples      (c) NF Samples      (d) HMC Samples

Figure 13: Another conditional inference example on CelebA dataset.

