# OpenReview forum: "Conditional Inference in Pre-trained Variational Autoencoders via Cross-coding"
_ICLR.cc/2019/Conference_

### Official Review · AnonReviewer1 · 2018-10-27
**Interesting read, but little original contribution**

**Rating:** 4
**Confidence:** 5

**Review:**

Paper summary:

Given a pre-trained VAE (e.g. over images), this paper is about inferring the distribution over missing variables (e.g. given half the pixels, what is a plausible completion?). The paper describes an approach based on variational inference with normalizing flows: given observed variables, the posterior over the VAE's latents is inferred (variationally) and plausible completions for missing variables are sampled from the VAE decoder.

Technical quality:

The presented method is technically correct. The evaluation carefully compares different types of normalizing flow and HMC, and seems to follow good practices.

I have a suggestion for improving the GVI method. The way it's described in the paper, GVI requires computing the determinant of a DxD matrix, which costs O(D^3), and there is no guarantee that the matrix is invertible. However, this approach over-parameterizes the covariance matrix of the modelled Gaussian. Without losing any flexibility, you can use a lower triangular matrix with strictly positive diagonal elements (e.g. the diagonal elements can be parameterized as the exp of unconstrained variables). That way, the determinant costs O(D) (it's just the product of diagonal elements) and you ensure that the matrix is invertible (because the determinant is strictly positive), without hurting expressivity. You can think of this as parameterizing the Cholesky decomposition of the covariance matrix.

Also, there are more flexible normalizing flows, such as Inverse Autoregressive Flow, that can be used instead of the planar flow used in the paper.

Clarity:

The paper is written clearly and in full detail, and the mathematical exposition is clear and precise.

Some typos and minor suggestions for improvement:
- It'd be good to move Alg. 1 and Fig. 1 near where they are first referenced.
- Page 2: over to \theta --> over \theta
- Eq. 3: p_\theta appears twice in the middle.
- one can use MCMC to attempt sampling --> one can use MCMC to sample
- Eq. 5: should be q_\psi as subscript of E.
- Fig. 7, caption: should be GVI vs. NF.
- In references, should be properly capitalized: Hamiltonian, Langevin, Monte Carlo, Bayes, BFGS
- Lemma 1: joint divergence is equivalent to --> joint divergence is equal to
- Lemma 1: in the chain rule for KL, the second KL term should be averaged w.r.t. its free variables.

Originality:

In my opinion, there is little original contribution in this paper. The inference method presented (variational inference with normalizing flows) is well-known and already in use. The paper applies this method to VAEs, which is a straightforward application of a well-known inference method to a relatively simple graphical model (z -> {x, y}, with x, y independent given z).

I don't see the need for introducing a new term (cross-coder). According to the paper, a cross-coder is precisely a normalizing flow (i.e. an invertible smooth transformation of a simple density). I think new terms for already existing ideas add cognitive load to the community, and are better avoided.

Significance:

In my opinion, constructing generative models that can handle arbitrary patterns of missing data is an important research direction. However, this is not exactly what the paper is about: the paper is about inference in a given generative model. Given that there is (in my opinion) no new methodology in the paper, I wouldn't consider this paper a significant contribution.

I would also suggest that in a future version of the paper there is more motivation (e.g. in the introduction) of why the problem the paper is concerned with (i.e. missing data in generative models) is significant. Is it just for image completion / data imputation, or are there other practical problems? Is it important as part of another method / solution to another problem?

Review summary:

Pros:
- Technically correct, gives full detail.
- Well and clearly written, precise with maths.
- Evaluation section interesting to read.

Cons:
- No original contribution.
- Could do a better job motivating the importance of the problem.

Minor points:
- I don't completely agree with the way VAEs are described in sec. 2.1. As written, it follows that VAEs must have a Gaussian prior and a conditionally independent decoder. Although these are common choices in practice, they are not necessary: for example, one could take the prior to be a Masked Autoregressive Flow and the decoder a PixelCNN.
- Same for observation 1. This is not an observation, but an assumption; that is, the paper assumes that the decoder is conditionally independent. This is of course an assumption that we can satisfy by design, but it's a design choice that restricts the decoder in a specific way.

---

### Official Review · AnonReviewer3 · 2018-11-02
**I don’t quite see what is new about this paper**

**Rating:** 4
**Confidence:** 4

**Review:**

This paper proposes the use of unamortized Black Box Variational Inference for data imputation (given a fixed VAE with a factorized decoder), where the choice of variational distribution is a standard flow model.

The exploitation of the decoder factorization and the choice to set q(y | z) = p(y | z) was explored in the Bottleneck Conditional Density Estimation paper.

To my understanding, this paper fails to contextualize their work with the existing literature and is simply an exercise in the rote application of existing inference procedures to a well-established inference problem (data imputation).

Unless the authors can convince me of the novelty of their approach or what I have overlooked in their proposal, I do not recommend this paper for acceptance.

References:
Ranganath, et al. Black Box Variational Inference. AISTATS 2014.
Shu, et al. Bottleneck Conditional Density Estimation. ICML 2017.

---

### Official Review · AnonReviewer2 · 2018-11-07
**A paper that needs work in terms of motivation, exposition, and evaluation**

**Rating:** 4
**Confidence:** 4

**Review:**

(apologies for this belated review)

Summary

The authors consider the task of imputing missing data using variational auto-encoders. To do so, they assume a fixed pre-trained generative model, perform variational inference to infer a posterior on latent variables given a partial image, and then use this approximate posterior to predict missing pixels. They compare a variety of parameterizations of the variational distribution to HMC inference, and evaluate on MNIST, Celeb-A and the Anime data.

Comments

There are many things about this paper that I don’t understand. My main concern is that I fail to follow why the authors are interested in this task. In what settings would we be interested in performing non-autoencoding variational inference in order to impute missing data? Moreover, in cases where are interested in performing such imputations, what would we like to use the results for? This paper seems like a nice demo, but I’m not entirely convinced I see a compelling application.

My second concern is about the baselines that are considered. If I were interested in carrying out this inference task, my inclination would not be to run an HMC chain to convergence, but instead to do something like annealed importance sampling (AIS), where at each step I run an iteration of HMC on a large batch of samples on a sequence of target densities that interpolate between the prior and full joint p(x, Z). If computational cost is a concern, I imagine this would not be more expensive than training a density estimator. Moreover, whereas HMC is generally not known to be a good method for estimating marginal likelihoods, AIS methods generally perform much better.

Finally I find the language used in this paper confusing. Cross-coding seems a misnomer for the technique that the authors propose. Isn’t this simply a form of variational inference in which qψ(Z) approximates pθ(Ζ | x)? The term “-coding” suggests that we somehow define an encoder that accepts the query as input. Moreover, isn’t the XCoder network just a neural density estimator?

Finally, Lemma 1 seems like a really roundabout way of deriving a lower bound. The authors could instead just write:

	log p(x)
	>=
	E_q(Z,Y)[log p(x, Y, Z) - log q(Z, Υ)]
	=
	E_q(Z,Y)[log p(x | Ζ) + log p(Y | Z) + log p(Z) - log q(Z) - log p(Υ | Z)]
	=
	E_q(Z,Y)[log p(x | Ζ) + log p(Z) - log q(Z)]
	=
	E_q(Z)[log p(x, Ζ) - log q(Z)]

This avoids confusing terminology such as cross-coding, and shows that what the authors are doing is in fact just variational inference. Am I missing something here?

I am also confused about how the comparison to HMC is set up. If you’re training qψ(Z), then you presumably need generate a certain number samples at training time. Shouldn’t you add this number of samples number of samples you generate in HMC, in order to get a more apples to apples comparison in terms of the amount of computation performed? As it stands, it is hard to evaluate whether these methods are given a similar number of samples.

Finally, I am not quite sure what to make of the experimental evaluation. We see some scatter plots on MNIST with a 2D latent space, and some faces of celebrities in which there is arguably some sample diversity, although most of this diversity arises in blurry looking hairstyles. However, since the authors condition on the eyes, rather than, say, the nose or mouth, it is hard to know how good a job the network is doing at generalizing to multiple plausible faces.

Overall, I find it difficult to judge the merit of this paper. Is this task in fact hard? Is it useful? Are the results good? Maybe the authors can give us some additional guidance on these questions.

Questions

- I’m a bit worried that not all the samples that we see in Figure 6 may have equally high probability under the posterior. Could the authors compute and report importance weights?

	W = p(x, Z) / q(Z)

- Could the authors say something about the effective sample size that we obtain when using the learned distribution q(Z) as a proposal?

	ESS = (Σ_k w^k)^2 / (Σ_k (w^k)^2)

- Should it be the case that the ESS is low, and the weights are high variance, could the authors generate a sufficient number of samples to ensure the the ESS = 25 (i.e. the number of images in the figure) and then show the 25 highest-weight samples (or resample 25 images with probability proportional to their weight)?


Minor


- Equation (3): There’s an extra p_θ in the first integral

- In the proof in Appendix 6.1

	KL[ qψ(Z) ‖ pθ(Z | x) ] + KL[ qψ(Y | Z) ‖ pθ(Y | Z, x)]

it would be clearer to explicitly denote the expectation over qψ(Z)

	KL[ qψ(Z) ‖ pθ(Z | x) ] + E_qψ(Z)[ KL[ qψ(Y | Z) ‖ pθ(Y | Z, x)] ]

(I had to google lecture notes to find out that this expectation is sometimes implicit, which
as far as I know is not very standard).

---

### Author Response · Authors · 2018-11-26
**Author Feedback**

Thanks for reviewers for their comments.

The goal of the paper is to infer p(y|x). The reviewers miss the central point of the paper which is using the framework of Augmented VI to understand why it is justifiable to do inference targeting p(z|x) and the looseness therefore entailed. Note that Rezende at al tackle EXACTLY this problem and design a custom algorithm, which we compared against. If it's all so simple, why would they even do that?

Reference:
Danilo Jimenez Rezende, Shakir Mohamed, and Daan Wierstra. Stochastic backpropagation and
approximate inference in deep generative models. In Proceedings of the 31th International
Conference on Machine Learning (ICML), pp. 1278–1286, 2014.

---

> ### Comment · AnonReviewer1 · 2018-11-28
> **Response**
>
> Thanks for your feedback.
>
> My score of 4 was mainly due to the lack of original contribution. The paper is technically sound, clearly written and interesting to read, but the inference methods discussed are already known and well-understood in the variational-inference community. There isn't anything in the authors' feedback that convinces me I missed something, so I'm afraid my review will remain the same.
>
> I sincerely hope that the authors find positive and constructive feedback in our reviews, and that they appreciate our good intentions and time we put in to help improve the paper. I wish the authors best of luck with their work.

---

### Meta-Review · Area_Chair1 · 2018-12-10
**Not well motivated and lack of novel contribution**

**Confidence:** 4
**Recommendation:** Reject

**Metareview:**

This paper proposes to approximate arbitrary conditional distribution of a pertained VAE using variational inferences. The paper is technically sound and clearly written. A few variants of the inference network are also compared and evaluated in experiments.

The main problems of the paper are as follows:
1. The motivation of training an inference network for a fixed decoder is not well explained.
2. The application of VI is standard, and offers limited novelty or significance of the proposed method.
3. The introduction of the new term cross-coding is not necessary and does not bring new insights than a standard VI method.

The authors argued in the feedback that the central contribution is using augmented VI to do conditioning inference, similar to Rezende at al, but didn't address reviewers' main concerns. I encourage the authors to incorporate the reviewers' comments in a future revision, and explain why this proposed method bring significant contribution to either address a real problem or improve VI methodology.